# Patterns of ASFV Transmission in Domestic Pigs in Serbia

**DOI:** 10.3390/pathogens12010149

**Published:** 2023-01-16

**Authors:** Dimitrije Glišić, Vesna Milićević, Ljubiša Veljović, Bojan Milovanović, Branislav Kureljušić, Igor Đorđević, Katarina Anđelković, Jelena Petković, Miroljub Dačić

**Affiliations:** 1Department of Virology, Institute of Veterinary Medicine of Serbia, 11000 Belgrade, Serbia; 2Department of Pathology, Institute of Veterinary Medicine of Serbia, 11000 Belgrade, Serbia; 3Department of Epizootiology, Veterinary Specialized Institute “Jagodina”, 35000 Jagodina, Serbia

**Keywords:** African swine fever, domestic pigs, wild boar, disease drivers, Serbia

## Abstract

The first case of African swine fever in domestic pigs in Serbia was in 2019. The following year, the disease was confirmed in wild boar. Thenceforth, ASF has been continuously reported in both wild and domestic pigs. The outbreaks in domestic pigs could not be linked directly to wild boars, even though wild boars were endemically infected, and reservoirs for ASF. This study aimed to investigate outbreaks and routes of transmission in domestic pigs in a region of central Serbia where no outbreaks in wild boar were reported. Fourteen outbreaks of ASF on backyard farms with low biosecurity were traced back, and no connection to wild boar was found. The epidemic investigation covered 2094 holdings, with 24,368 pigs, out of which 1882 were tested for ASF. In surrounding hunting grounds, field searches were conducted. Dead wild boars were found, and 138 hunted wild boars were negative for ASFV. It was concluded that outbreaks in 2021 were provoked by the illegal trade of live animals and pig products. Even though infective pressure from wild boars is assumed, no positive cases have been found, while the ASFV spreads within the domestic swine population evidenced in four recent outbreaks in 2022.

## 1. Introduction 

African swine fever (ASF) is an infectious viral disease with a high case fatality in domestic pigs and wild boar. When the disease appears, a high socio-economic impact is experienced by the farmers, affecting the farmers’ income, and concurrently affecting regional and international trade. Since there is still neither an effective treatment nor an effective vaccine [1], strict biosecurity measures have to be applied in order to prevent ASF in domestic pigs, whereas, in the wild boar population, targeted hunting, removal of wild boar carcasses in the wild, and a strict feeding ban are the main control measures [2]. The causal agent is the African Swine Fever Virus (ASFV) which belongs to the Family *Asfaviridae* [3]. It is one of the most complex DNA viruses known [4]. The resistance and stability of ASFV are considered the major elements of disease maintenance: ASFV survives different environmental conditions, including the meat curing process, and remains infectious over a long period at temperatures below 4 °C [5]. Similarly, the virus survives in different types of soil depending on pH, organic material, and temperature [6,7]. During the last 15 years, since ASFV was brought from Africa to Eurasia, the disease spread throughout Asia, Europe, the Caribbean, and the Pacific, gaining characteristics of an endemic disease, with recorded survival animals [8]. In these regions, transmission cycles of ASFV differ substantially from the virus transmission in Africa, where ASFV originates from and where soft ticks of the family *Argasidae*, genus *Ornithodoros*, and African wild suids are identified as biological vectors and reservoirs [9]. Three recognized factors influence ASF occurrence: the host, the landscape, and the factors related to human activities [10]. Out of Africa, swill-feeding and pig/pork transport are considered the main transmission routes, whereas direct contact between infected and non-infected animals has a limited role in long-distance transmission [11]. 

In Uganda [12], as well as in rural areas of China, there have been reports of “panic selling” of pigs, and pig products, due to lower prices offered by the producers within infected areas. Similarly, it can be noticed in Serbia, where the lack of belief in government reimbursement, or the possibility of making a “quick turnaround” expedites the selling of apparently healthy pigs. The term “amplifying spots of ASFV” denotes the places where individual farmers, trying to limit economic losses, either emergency sell or slaughter pigs for consumption when the disease occurs, thereby enabling the rapid transmission of ASFV. However, for a better understanding and exploration of the more effective prophylactic and reactive management actions against ASFV, an individual-based approach is recommended to comprehensively investigate and consider all the local patterns of the disease, including characteristic outbreaks in wild boars, potential amplifying spots, virus sources, and other potential drivers such as habits of the human population, and vectors. It also includes ASFV phylogeny and phylogeography, which provide information on its diversification, ecology, and evolution. ASFV genotypes appear to be very homogeneous, and only in some sub-genotypes can they be further differentiated [13]., exemplified by the existence of 24 genotypes and many different subgenotypes in Africa, although only slight mutations have been noted within genotype II currently circulating through the world [14,15,16]. Since ASF was present in the wild boar population in Serbia since January 2021, and given the complexity of ASF, the main aim of this study was to determine the most common infection sources and transmission routes in conjunction with local virus strain characteristics and biosecurity assessment to enable adjustment of the disease control measures. In particular, humans’ role as the main vector in ASFV transmission was addressed.

## 2. Materials and Methods 

The data used for this study are descriptive epidemiological data that were generated during the outbreak investigations. The epi-investigations included the determination of the High-Risk Period (HRP), exact geographical locations of the outbreak (GPS coordinates), names and addresses of affected farmers/farms, the numbers of sick and dead pigs, approximate numbers of susceptible animals in the area, brief descriptions of clinical signs and pathological lesions, the date when ASF was first noticed, details of recent movements of pigs to or from the outbreak farm, details of any recent movement of trucks and people from or towards other farms, number of workers at the farm, frequency of feeding, and other epidemiological information such as the presence of the disease in wild boars and abnormal insect activity. For the biosecurity assessment, both external and internal, the biosecurity scoring system BiocheckUGent™ (https://biocheckgent.com/en accessed on 4 November 2022) was applied for 10 backyard farms that were the most common type of farm in this region. Each assessed category was scored in a rank from 0 (the worst scenario) to 100 (the best scenario). The overall biosecurity was computed as the average of external and internal biosecurity scores. The post-epidemic investigation included 2094 holdings, with 24,368 animals, out of which 1882 were tested for ASF. In addition to domestic pigs, field searches for dead wild boar were conducted in surrounding hunting grounds. Data relating to local human awareness toward ASF, knowledge about ASF clinical signs, presence in the country/district, ways of spread, and the readiness of the owners to report suspected cases of ASF infection was collected through surveys. In total, 137 surveys were conducted (Appendix A). Microsoft Excel software was used for data analysis, epidemiological statistical calculations, and visualisations (version 2019, Microsoft, Redmond, WA, USA). Google maps were used for generating the epidemiological maps (Map 1.). For the detection of the ASFV genome, a real-time PCR protocol using primers previously described by King et al. (2003) [17], which amplifies the VP72 protein gene, was used. The master mix included 10 μL of Luna Universal Probe qPCR Master Mix (Luna® Universal Probe qPCR Master Mix, New England Biolabs, Ipswich, MA, USA), 0.8 μL of each 10 mM primer, 0.4 μL of the 10 mM probe, and 2 μL of DNA template, the rest up to 20 μL was supplemented with RNA free water. The temperature profile used included initial denaturation at 95 °C for one minute, followed by 50 cycles of 95 °C for 15 s, and 60 °C for 30 s. Samples deemed positive were further genotyped. For genotyping, two different sets of primers were used. For the determination of the genotype, a B646L gene coding the VP72 protein was targeted, by primers previously described by Bastos et al. (2003) [18]. For further in-depth sequencing, the central variable region (CVR) region within the B602L gene was targeted, with primers previously described by Gallardo et al. (2009) [19]. For both reactions, 10 μL of HotStar Master Mix (QIAGEN HotStarTaq Master Mix, Les Ulis, France), 0.6 μL of each 10 mM primer were used, and 2 μL of DNA template used, the rest up to 20 μL was supplemented by RNA-free water. The thermal profile included 95 °C for 15 min, followed by 40 cycles of 95 °C for 30 s, 50 °C for 30 s for the VP72 gene, and 55 °C for one minute for the B602L gene, followed by 72 °C for one minute, and the final extension at 72 °C for 10 minutes. The amplified PCR products (478–500 bp) were visualised by electrophoresis on 1.5% agarose gel stained by ethidium bromide. PCR products were purified using GeneJET PCR Purification Kit (ThermoFisher Scientific, Waltham, MA, USA) and sequenced at LGC, Biosearch Technologies, Germany, by the Sanger sequencing method. In total, 16 sequences were used for the phylogenetic study, including 10 sequences obtained from domestic pig outbreaks (Pomoravlje region), and 6 sequences originating from wild boar from the neighbouring infected regions. The consensus sequences were generated in Geneious Prime (Geneious Prime, Dotmatics, Boston, MA, USA) software. Sequences of the B646L gene were trimmed to the length of 297 bp and were aligned with 27 strains from National Center for Biotechnology Information (NCBI) (Appendix A). Whereas, sequences of the B602L gene were trimmed to the length of 302 bp and were aligned with 20 sequences from the NCBI (Appendix A). For the construction of both the phylogenetic trees, the Molecular Evolutionary Genetic Analysis (MEGA X) software was used. The phylogenetic trees were constructed by using the Neighbour-Joining Method, and the Jukes–Cantor model, with 1000 bootstrap replicates, and uniform rates among sites.

## 3. Results 

During 2021–2022, fourteen ASF outbreaks were reported in the central Pomoravlje region in Serbia. ASF was first noticed on 2 March 2021, and followed by six outbreaks in four settlements on the territory of Paraćin municipality (Figure 1). Following the report of clinical signs in the first backyard farm in Paraćin municipality, the same veterinary service member made visits to other six backyard farms, in which ASF was later recorded simultaneously. The settlements are of the nucleated type, and farms of the backyard type. In 2022, ASF continued to spread and seven outbreaks were recorded. In the municipality of Despotovac, ASF occurred in two waves, the spring, and the autumn wave, in three settlements, whilst in Ćuprija, two outbreaks in free-range farms were recorded in one dispersed settlement. The average number of pigs per farm was 15.

To contain ASF, 238 pigs were euthanised in this region. Average mortality was 8.81%, while lethality reached 100%. The estimated high-risk period was defined as 21 days for backyard farms, and 35 days for free-range farms. The most affected category were gestating sows, before farrowing, or just after. In only one farm, fattening pigs were affected. The total number of susceptible animals in the area was 19,666 animals. Clinical signs were characteristic of the peracute and acute course of the disease. The disease lasted up to three days on average, with a high fever of 40–41 °C, depression, anorexia, weakness, abortions, and haemorrhages in the skin, particularly on the ears. The common pathological lesion were hemorrhagic lymph nodes, splenomegaly, renal cortical petechiae, pulmonary oedema, and hemopericardium. 

Other important epidemiological data are that swill feeding was rarely practised, but natural mating was common. None of the owners were hunters, but contacts with hunters were common. The intensive trade of animals was recorded before the first outbreak in 2021. The average number of workers at the farms was 2.5, and were primarily the owners’ family members. The frequency of feeding was 2–3 times a day or ad libitum in free-range farms. The disease was not confirmed in a wild boar in a radius of 60 km at the time of the first outbreak. However, the closest outbreak in domestic pigs was recorded in January 2021, 30 km south, along the main road (Figure 1). 

The results of the biosecurity assessment are shown in Table 1. 

The overall biosecurity score was 27.6 ± 2.07. The results of the external biosecurity assessment showed a mean score of 20.2%. External biosecurity assessment showed the lowest scores for the transport of animals, carcasses, and manure disposal (8.1%). The assessment of internal biosecurity revealed a score of 30.2% with the lowest percentage score (0%) for cleaning and disinfection performed. In surveys taken, it was found that the general populous was aware of existing ASF cases in neighboring municipalities, but that they were not sure what their role in disease management should be, what was expected of them, and there was a belief that a systemic strategic state-wide approach is necessary. Furthermore, with a lack of a unified approach, a certain number of people have expressed doubt about the very existence of the disease, dismissing it as a hoax. Post-outbreak investigation revealed that none of the domestic pigs and wild boars were found to be ASFV positive in the area. The results of the phylogenetic analysis of the B646L gene revealed that all strains examined in the study belong to genotype II of ASF, and no differences were observed between strains obtained from domestic pigs in the Pomoravlje region and strains obtained from wild boars from the surrounding region (Figure 2). The phylogenetic analysis of the B602L gene revealed that all strains used in this study belong to the CVR 1 subtype (Figure 3). 

## 4. Discussion 

ASF is a devastating disease that most heavily affects small producers. The “human factor” in disease transmission includes low biosecurity measures, and low owner awareness stemming from the lack of education regarding the disease transmission and its socio-economic impact. In this study, we traced back outbreaks in the backyard and free-range farms to define routes of transmission and the high-risk activities that contributed to virus spread. Concerns regarding the eradication success of ASF in domestic pigs persist, as wild boar remains the main reservoir of the disease [20]. In order to determine the possible self-sustainability of ASF within the wild boar population, cases of survivors are necessary. Since the first occurrence of the disease within the wild boar population, an active surveillance program for the detection of antibodies against ASFV has been conducted. The prevalence of ASF in wild boar remains difficult to assess since the estimation depends on passive surveillance, and the reporting of wild boar carcasses, usually by hunters or foresters. In a study by Vergne et al. [21], conducted in Bulgaria, Germany, and the Russian Federation, 70–80% of hunters that came across a wild boar carcass have reported it, while those that have not cited a “lack of awareness” as the main reason. Other possible reasons for not reporting, not citing, could include the possible closure of hunting grounds, and as such, a lapse in profits. However, in this work, we showed that outbreaks and eradication processes in domestic pigs in poor, low-biosecurity settings are not dependent on ASF occurrence in wild boar. A similar case was shown in Sardinia where ASF in wild boar could not have been maintained without a continuous source of the virus in illegal free-ranging domestic pigs [22]. Such persistence factors in Serbia are based on illegal movement, and selling of animals, or the illegal slaughter of pigs suspected of ASF. Since ASF was introduced in Serbia in 2019 [23], it has continuously been reported in backyard farms and wild boars making it as endemic as in Romania and Bulgaria. Evidence regarding the involvement of wild boar in African swine fever transmission in Serbia is scarce and prompts further investigation. The index case of ASF in Serbia was in July of 2019, as a result of imported illegal pig products from affected countries, the second outbreak followed less than a month later, also connected with human activities. Both cases were resolved by a swift stamping out policy, and no new cases were recorded until the next year. Later on, in January 2020, ASF was naturally introduced from Bulgaria following spatial continuity in the direction E-W [24]. This event was expected as per the calculated speed of propagation of ASF [24,25]. In 2021, ASF occurred in a large commercial farm keeping 19,000 pigs in the infected area. Though the highest biosecurity measures were implemented, Nešković et al. [26] showed that the virus entered that farm via different anthropogenic activities. Control of classical swine fever which is a highly contagious disease compared to ASF is a good example that strict biosecurity measures can prevent virus spillover from wild boar to domestic pigs, even when CSF vaccination is not practised in domestic pigs [27,28]. All 14 outbreaks that were traced back in the central Pomoravlje region occurred in the backyard and free-range farms. Almost 70% of holdings in Serbia keep up to 10 pigs, indicating that backyard holdings are very common and traditional in the rural areas of Serbia [29]. Commonly, backyard pigs are slaughtered at the farm, and used for their consumption. In addition, the products are usually shared with family and friends. Though there is a ban on swill feeding, it cannot be excluded. Costard et al. [30] showed that small producers present a high risk of virus spreading. Despite educating farmers to recognize the disease, they are reluctant to report it to authorities, and would rather either sell pigs without clinical signs or slaughter pigs with clinical signs [30]. This means that ASF eradication and control rely on farmers‘ awareness. Thus, awareness and biosecurity measures that small farmers implemented should be considered. The results of the biosecurity assessment conducted on 10 farms in the affected area confirm that backyard farm owners were not aware of the importance of biosecurity measures. The averages at backyard farms were significantly lower than the average values reported in Europe [31]. In particular, the highest risk comes from not performing disinfection and cleaning as none of the farms use it routinely. In some rural areas, there are cases where pigs in pens are not attended to, and feed is only added until the pigs reach the finishing weight for slaughter. Afterwards, the pens are mechanically cleaned and washed, without the use of disinfectants before new pigs are introduced. The lowest scores were recorded regarding external biosecurity, the transport of animals, removal of manure, and disposal of carcasses. Carcass disposal is regulated by law, although the monetary burden falls on the owner, and thus in rural areas carcasses are often discarded, or the death of an animal is not reported. This was the case in Paraćin where dead pigs were found disposed of by the regional road. However, such attitudes toward biosecurity, although contradictory at first glance, can be understood to be taking into consideration the poor economic disposition of such households, where backyard pig farms represent the main income for their family. In such occasions, the lack of monetary support from the government, or even governmental prosecution for lacking adequate biosecurity measures where none are financially manageable for the farm owner, could be a contributing factor to concealing diseased and/or dead pigs in infected areas. The mentioned hypothesis does not alleviate the owner’s responsibility to establish some biosecurity measures (disinfection barriers, disinfection of footwear, and limiting the number of visitors, etc.), primarily the application of disinfectants. Other authors highlight that human behaviors and activities present the highest risk for virus spread. Even in South Africa, the roles of wild pigs and competent vectors are considered relatively minimal compared to human factors [32]. Trades, transport, farm visitors, and behaviour by farm personnel are defined as human-related drivers of most concern [33,34,35]. 

Considering HRP is estimated at 21 days for backyard farms and 35 days for free-range farms, and different activities on the farm, the risk of spreading the disease multiplies. 

The Pomoravlje region in 2021 was not in a restricted area, free from ASF in both domestic and wild pigs. However, there were outbreaks reported in the bordering municipalities in domestic pigs in January 2021. This area is famous for swine production and livestock markets that are organized every week. After the ASF outbreaks in the wider area were confirmed and control measures applied, an intensive trade at reduced prices was recorded in the Pomoravlje region. Thus, the trade of infected animals, pig products, and indirect routes by fomites were the most likely source of infection in the free area. 

In 2022, a part of Pomoravlje was in an endangered region, established according to ASF outbreaks in wild boars in the bordering municipality. The density of the wild boar population in this area is estimated at 0.5–1 wild boar/km^2^, which is relatively low compared to other European areas [36]. Though ASF was confirmed to persist even in low-dense populations [37], the complexity of ASF and efficient indirect transmission indicates that human activities were the more likely vector of the virus. Furthermore, ASF in wild boar was neither found in the post-outbreak investigation nor later on. 

Additionally to intensive trade at low prices that usually peaks in case of disease occurrence that enables long jumps of the disease, along with inefficient trade control our results suggest iatrogenic transmission contributes to the spread of the disease within a settlement and holding. The possibility of contaminated swill feeding, despite the prohibition, cannot be excluded as a possible way of disease transmission within the farm, whereas in the settlement, the iatrogenic transmission was the main transmission route. Considering that ASF is not a highly contagious disease, and based on the time of disease occurrence, in 2022 the same source of infection was defined. Unfortunately, this indicates that awareness and education are a risk not only for farmers but to veterinary professionals too. The results from this study need to be interpreted taking into account the limitations of descriptive epidemiology and the accuracy of information obtained by interviewing. Thus, we assume that the real situation can only be more serious and complex. In such cases, molecular epidemiology could be used for clarification and confirmation, but the ASFV genome was shown to be stable, thus making phylogenetic tracing very difficult. The stability of genotype II of the ASFV, coupled with the length and complexity of the genome make in-depth whole-genome sequencing very laborious [38] and often not cost-effective, especially for large-scale analysis, and also very difficult to achieve for low-income countries. Furthermore, Sanger sequencing methods although more economical, have issues in plurality, where six different genomic markers need to be sequenced to obtain enough information regarding the epidemiological origin of the viral strain, which is sometimes not feasible. In this study, a B602L gene was sequenced, which offers differentiation into 31 different strains, although only 2 strains for genotype II of the ASFV, of which the CVR 2 strain, have only been detected once in Estonia [39]. This necessitates a more targeted approach for the phylogenetic analysis, primarily based on the clinical aspects of the disease observed by field veterinarians. Even though all samples in this study belong within genotype II CVR I of ASFV, there may be mutations at different parts of the genome, and as such should be further investigated.

Although ASF is widespread in wild boar in Serbia, in this research region it can be observed that one of the main disease drivers is human action. There is a severe risk that backyard farms will become a continuous source of ASFV for domestic and wild boar and that new outbreaks in the wider area can be expected unless more efficient measures are applied. Education and awareness campaigns must be continuously organized. The human stands for the main risk of disease spreading. 

## Figures and Tables

**Figure 1 pathogens-12-00149-f001:**
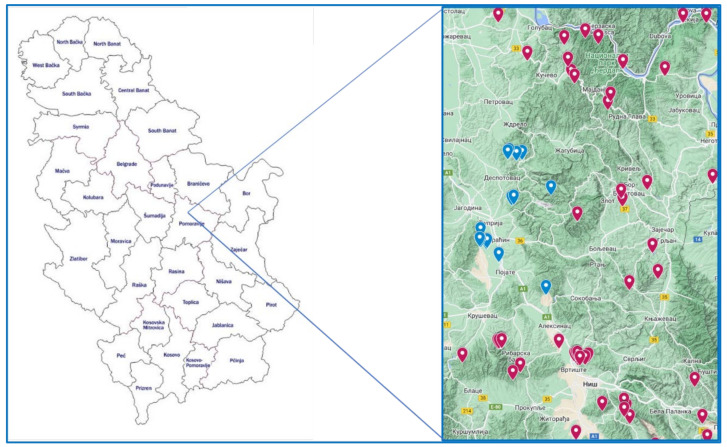
Map of Serbia, with the Pomoravlje region, and the surrounding municipalities where ASF-positive cases were recorded, generated with Google MyMaps (USA). Blue dots represent the cases within the Pomoravlje region, and the red dots, cases of positive wild boar in the surrounding municipalities. The red dots represent all positive cases in wild boar from November 2021 up to October 2022 [19]. The blue circle represents the isolated cases of domestic pigs in the Pomoravlje region.

**Figure 2 pathogens-12-00149-f002:**
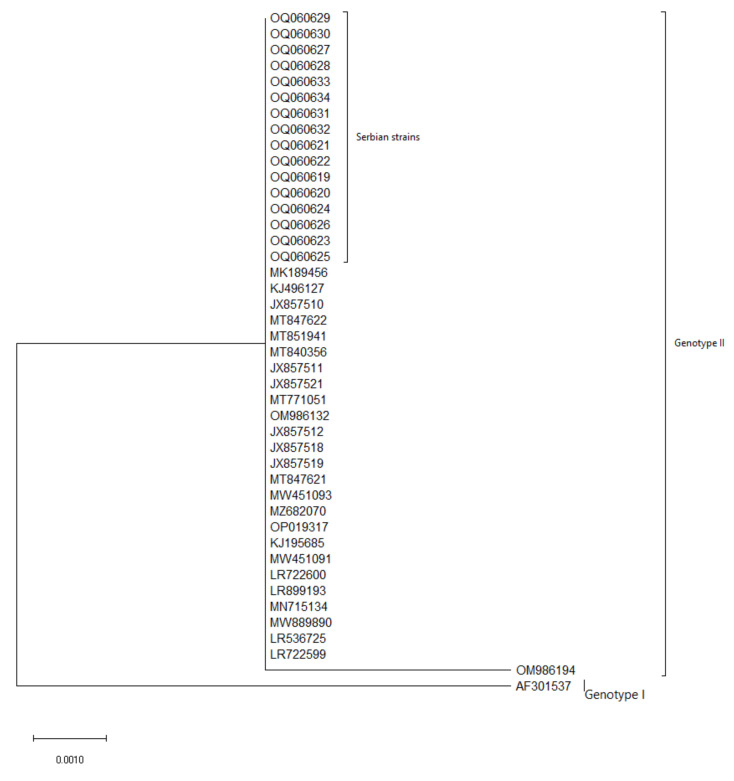
Phylogenetic tree illustrating the genetic relationships between sequences of the B646L gene of Serbian strains (OQ060619 – OQ060634) and strains obtained from the NCBI (Appendix A). Phylogenetic analysis was conducted in MEGA X software using the Neighbour-Joining Method and the Jukes–Cantor model, with 1000 bootstrap replicates and uniform rates among sites. Branches corresponding to partitions reproduced in less than 70% of bootstrap replicates were collapsed. One out-of-group sequence of genotype I ASFV was added for comparison (AF301537).

**Figure 3 pathogens-12-00149-f003:**
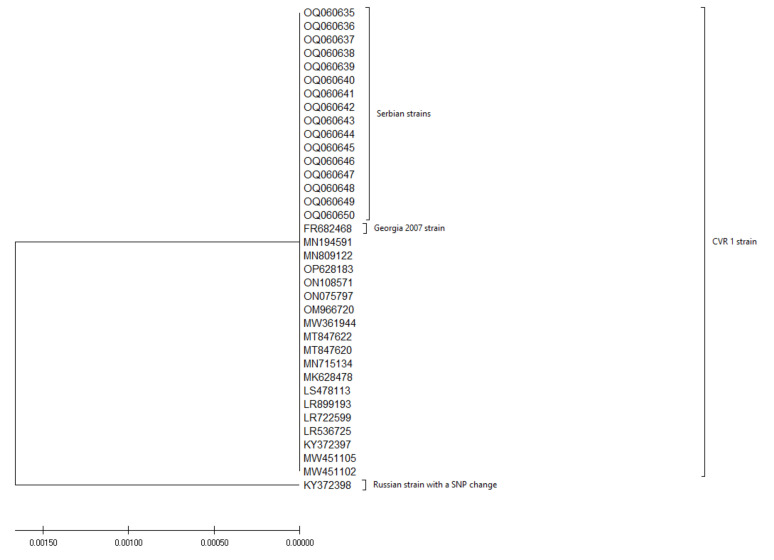
Phylogenetic tree illustrating the genetic relationships between sequences of the B602L gene of Serbian strains (OQ060635 – OQ060650) and strains obtained from the NCBI (Appendix A). Phylogenetic analysis was conducted in MEGA X software using the Neighbour-Joining Method, and the Jukes–Cantor model, with 1000 bootstrap replicates, and uniform rates among sites. Branches corresponding to partitions reproduced in less than 70% of bootstrap replicates were collapsed. One out-of-group sequence with an SNP mutation was added for comparison (KY372398).

**Table 1 pathogens-12-00149-t001:** Biosecurity scores (BiocheckUGent™) for the different categories of internal and external biosecurity in 10 pig holdings.

	Mean	SD	Median	Min	Max
External biosecurity score	20.2	2.39	20	16	24
Purchase of animals and semen	29.7	4.52	30	22	38
Transport of animals, removal of manure and dead animals	8.1	1.45	8	6	10
Feed, water, and equipment supply	24	3.27	24	20	30
Personnel and visitors	8.8	2.44	8.5	6	14
Vermin and bird control	17.6	2.27	17.5	15	22
Location of the farm	28.4	3.20	28.5	24	35
Internal biosecurity score	30.2	1.32	30	28	32
Disease management	18.8	2.04	19	16	22
Farrowing and suckling period management	63	2.58	62.5	60	68
Nursery unit management	51.1	3.54	51	45	56
Fattening unit management	50.4	3.20	50.5	45	55
Measures between compartments and use of equipment	21.7	2.50	21.5	18	25
Cleaning and disinfection	0	0.00	0	0	0
Overall biosecurity score	27.6	2.07	28	24	30

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
