# Peer review of "Patterns of ASFV Transmission in Domestic Pigs in Serbia"

_pathogens, 2023, doi:10.3390/pathogens12010149_

Round 1

Reviewer 1 Report

This manuscript is descriptive of various parameters that are assumed to play a part in the spread and persistence of ASFV in low biosecurity swine producing units. The conclusions are relevant to not only the authorities in the locations under investigation but also important to emergency response planners in non-infected regions. 

There are a few word substitutions highlighted in the attached file that the authors should consider. 

Author Response

Dear,

We would like to thank the reviewers for their comments, and for the time and expertise that they offered.

Below you can find our responses to the reviewers’ comments in three sections, one for each reviewer. The lines mentioned in the responses relate to the updated manuscript.

In general two tables were added as supplementary material, Tables 1 and 2 which concern a questionnaire and answers to the questionnaire. Other tables in the supplementary materials have been renamed accordingly.

Changes made to the supplementary material and the Data Availability Statement

Line 328-333: Supplementary Materials: Table 1. S1. Title, List of questions. Table 2. S2. Title, Answers to the questionnaire. Table 3. S3. Title, Sequences used in the B646L gene phylogenetic study. The accession numbers of the B646L gene from the NCBI were used for the alignment with sequences from this study. Table 4. S4. Title, Sequences used in the B602L gene phylogenetic study. The accession numbers of the B602L gene from the NCBI were used for the alignment with sequences from this study.

Data Availability Statement: The data presented in this study are available in [Figure 1. S1., Figure 2 S2. Figure 3. S3, Table 1. S1, Table 2. S2, Table 3. S3., Table 4. S4, Table 5. S5.]

Point 1:

Line 36: Since there is still neither an effective treatment nor an effective vaccine. Strict biosecurity measures have to be applied in order to prevent ASF in domestic pigs, whereas, in the wild boar population, targeted hunting, removal of wild boar carcasses in the wild, and a strict feeding ban are the main control measures.

Response 1: We agree with the comment made by reviewer 1, and have made the necessary changes.

Line 36:Since there is still neither an effective treatment nor an effective vaccine, strict biosecurity measures have to be applied in order to prevent ASF in domestic pigs, whereas, in the wild boar population, targeted hunting, removal of wild boar carcasses in the wild, and a strict feeding ban are the main control measures.

Point 2: However, for a better understanding and exploring the more effective prophylactic and reactive management actions against ASFV, an individual-based approach is recommended, comprehensively investigating and considering all local patterns of the disease, including the character of outbreaks in wild boars, potential amplifying spots, virus sources, other potential drivers such as habits of the human population, and vectors.

Response 2: We agree with the comment made by reviewer 1, and have made the necessary changes.

Line 60: However, for a better understanding and exploring the more effective prophylactic and reactive management actions against ASFV, an individual-based approach is recommended, comprehensively investigating and considering all local patterns of the disease, including the characteristic of outbreaks in wild boars, potential amplifying spots, virus sources, other potential drivers such as habits of the human population, and vectors.

Point 3:

Line 88: Further to domestic pigs, field searches for dead wild boar were conducted in surrounding hunting grounds.

Response 3: We agree with the comment made by reviewer 1, and have made the necessary changes.

Line 93: In addition to domestic pigs, field searches for dead wild boar were conducted in surrounding hunting grounds.

Point 4:

Line 88: Data on the human attitude toward ASF awareness, knowledge about ASF clinical signs, presence in the country/district, ways of spread, and the readiness of the owners to report suspected cases of ASF infection was collected through surveys.

Response 4: We agree with the comment made by reviewer 1, and have made the necessary changes.

Line 95: Data relating to local human awareness toward ASF, knowledge about ASF clinical signs, presence in the country/district, ways of spread, and the readiness of the owners to report suspected cases of ASF infection was collected through surveys

Point 5:

Line 141: Clinical signs were characteristic of the peracute and acute course.

Response 5: We agree with the comment made by reviewer 1, and have made the necessary changes.

Line 149: Clinical signs were characteristic of the peracute and acute course of the disease.

Point 6:

Line 175:

The results of the phylogenetic analysis of the B646L gene reviled that all strains used in the study belong to genotype II of ASF, and no differences were observed between strains obtained from domestic pigs in the Pomoravlje region and strains obtained from wild boars from the surrounding region.

Response 6: We agree with the comment made by reviewer 1, and have made the necessary changes.

Line 187:

The results of the phylogenetic analysis of the B646L gene revealed that all strains examined in the study belong to genotype II of ASF, and no differences were observed between strains obtained from domestic pigs in the Pomoravlje region and strains obtained from wild boars from the surrounding region.

Point 7:

Line 266:

Additional to intensive trade at low prices that usually records in case of disease occurrence that enables long jumps of the disease, and along with inefficient trade control, our results suggest iatrogenic transmission contributes to the spread of the disease within a settlement and holding.

Response 7: We agree with the comment made by reviewer 1, and have made the necessary changes.

Line 301:

Additional to intensive trade at low prices that usually peaks in case of disease occurrence that enables long jumps of the disease, and along with inefficient trade control, our results suggest iatrogenic transmission contributes to the spread of the disease within a settlement and holding.

Point 8:

Line 277:

The results from this study need to be interpreted taking into account the limitations of descriptive epidemiology and the accuracy of obtained information by interviewing.

Response 8: We agree with the comment made by reviewer 1, and have made the necessary changes.

Line 307:The results from this study need to be interpreted taking into account the limitations of descriptive epidemiology and the accuracy of information obtained by interviewing.

Reviewer 2 Report

The manuscript describes the main routes of infection with ASFV in domestic pigs in an infected region in Serbia. The variables taken into consideration such as the level of implemented biosecurity measures in farms within the infected area and possibility of wild boar/domestic pig spillover give advance views of ASF epidemiology characteristics in Europe, and indicate that implementation of biosecurity measures may highly prevent the spread of ASF within an area, even postinfection. Therefore, education and awareness of farmers and other involved subjects is crucial for preventing and eradicating the disease.

However, please splell check the English. Minor mistakes are found.

Please use weeks OR days for the description of HRP, not both.

Author Response

Dear,

We would like to thank the reviewers for their comments, and for the time and expertise that they offered.

Below you can find our responses to the reviewers’ comments in three sections, one for each reviewer. The lines mentioned in the responses relate to the updated manuscript.

In general two tables were added as supplementary material, Tables 1 and 2 which concern a questionnaire and answers to the questionnaire. Other tables in the supplementary materials have been renamed accordingly.

Changes made to the supplementary material and the Data Availability Statement

Line 328-333: Supplementary Materials: Table 1. S1. Title, List of questions. Table 2. S2. Title, Answers to the questionnaire. Table 3. S3. Title, Sequences used in the B646L gene phylogenetic study. The accession numbers of the B646L gene from the NCBI were used for the alignment with sequences from this study. Table 4. S4. Title, Sequences used in the B602L gene phylogenetic study. The accession numbers of the B602L gene from the NCBI were used for the alignment with sequences from this study.

Data Availability Statement: The data presented in this study are available in [Figure 1. S1., Figure 2 S2. Figure 3. S3, Table 1. S1, Table 2. S2, Table 3. S3., Table 4. S4, Table 5. S5.]

Point 1: However, please splell check the English. Minor mistakes are found.

Response 1: We agree with the comment made by reviewer 2, and have made the necessary changes. Spelling mistakes were located and changed accordingly.

Point 2: Please use weeks OR days for the description of HRP, not both.

Response 2: We agree with the comment made by reviewer 2, and have made the necessary changes.

Reviewer 3 Report

In general, the topic of this paper is interesting and provides information about a very important and topical disease. Furthermore, it describes a context in which African Swine Fever is present, but in an underestimated form, and therefore the paper is of relevant interest to the scientific community.

The paper defines itself as being based on descriptive epidemiology methods, but it not only provides data on the context in which a number of ASF outbreaks have been reported, in fact it reports information on the genetic analysis of isolated strains and the results of a series of interviews without however clarifying the categories of people involved.

The main shortcoming I found in the paper is that the authors do not seem to give enough consideration to the epidemiology of wild boar in the context analysed. The paper reports that no cases of the disease in wild boar have been reported in the area in which the considered outbreaks are included, but does not consider possible epidemiological links between the observed outbreaks (in domestic pig farms) and the presence of the infection in wild boar populations in other (not too far away) areas of Serbia. Furthermore, it is stated that ASF in Serbia would not be self-sustainable in wild boars and that, as a consequence, the main factor of virus transmission is human-related. I might agree with the latter hypothesis, but the evidence regarding a secondary role of wild boar in the epidemiology of ASF in Serbia is scarce and should be further investigated.

Apart from these general considerations, I list some more specific observations below:

Title: the title refers to a well-known concept and is not entirely in keeping with the objectives described at the end of the introduction (lines 67-71). In practice, the title is commonplace and on the other hand, it is not clear why wild boar in Serbia should not already be considered a reservoir of ASF virus infection, and likewise it is not surprising that the human factor plays a key role. In addition, there is no reference in the title to phylogenetic analysis, which is instead mentioned among the aims of the work. In conclusion I suggest to modify the title

Introduction

Line 60-61; The concept 'amplifying spots' is interesting, but appears rather suddenly in the text and is not supported by any bibliographical references. It would be appropriate for the authors to elaborate on why this topic is mentioned and why it should be taken into account in relation to other factors favouring the spread of infection. In short, it would be a case of contextualising the concept of 'amplifying spots'.

Line 65-66; the statement 'the virus is genetically unstable' appears contradictory to what is stated later. In general, this virus cannot be defined as unstable. This issue needs to be reworked.

Materials and methods

Line 92; how many surveys were carried out? How were these surveys structured? At least a list of the submitted questions should be provided (supplementary material)

Results

The map used in figure 1 is rather small. It would also be useful to provide a map showing cases of ASF in wild boars.

The results of the interviews conducted are rather confusing: it would be appropriate to detail them using the list of questions requested above.

Discussion

Lines 202-204; the concept expressed is quite peremptory and perhaps too assertive. No consideration is given to the possibility that there may have been unreported cases in wild boar in the area where the outbreaks under study were recorded. This is an aspect that should be considered in the work. The potential underestimation of the prevalence and incidence of ASF in wild boar could be a determining factor affecting the whole epidemiological context considered. This topic should also be considered in the preceding chapters: a way should be found to analyse the distribution of cases in wild boars and the results should be shown in order to address this issue and possible connections with the disease in domestic animals. Furthermore, one question remains pending: is it possible that the disease in wild boars could still be self-sustainable in other zones of Serbia?

Line 204-206; the situation in Sardinia is more complex than described. The paper cited in the bibliography and other works identify the presence of a key persistence factor linked to the illegal free ranging pigs that are widespread in some areas of Sardinia; in this sense, the differences with Serbia are very marked and this aspect should be further investigated. The real question is: is it possible to hypothesise a key persistence factor (still unconsidered or underestimated) that conditions the spread and persistence of the ASF virus in Serbia in wild boars as in domestic animals?

Line 206-208; considering the above, the disease in Serbia seems more comparable to that in Bulgaria or Romania rather than Sardinia. Please change this sentence.

Line 242 244; I think this data is not contradictory: the problem is an economical  one, and this is important to note and address in the discussion. If farmers are poor and few pigs are their only source of livelihood, it is very likely that they do not have the money to improve the biosecurity of their herds; this could explain the tendency to hide any health problems.

Line 268; you state that 'our results suggest iatrogenic transmission': what results are you referring to? What data support this statement? Honestly, I found no evidence in the paper

Line 277; "The results from this study need to be interpreted taking into account the limitations .... and the accuracy of obtained information by interviewing"; this is why the number of interviews and the list of questions are requested

Line 278-280; the discussion regarding genetic analysis is too narrow and limited to these few words. The topic merits further discussion and could be linked both to the transmission of the virus between the domestic and the wild and, more generally, to the concept of virus stability

Round 2

Reviewer 3 Report

Given the revisions the authors have applied to the paper, I believe it is ready to be released.